# pEBR: A Probabilistic Approach to Embedding Based Retrieval

## Abstract

Embedding retrieval aims to learn a shared semantic representation space for both queries and items, thus enabling efficient and effective item retrieval using approximate nearest neighbor (ANN) algorithms. In current industrial practice, retrieval systems typically retrieve a fixed number of items for different queries, which actually leads to insufficient retrieval (low recall) for head queries and irrelevant retrieval (low precision) for tail queries. Mostly due to the trend of frequentist approach to loss function designs, till now there is no satisfactory solution to holistically address this challenge in the industry. In this paper, we move away from the frequentist approach, and take a novel **p**robabilistic approach to **e**mbedding **b**ased **r**etrieval (namely **pEBR**) by learning the item distribution for different queries, which enables a dynamic cosine similarity threshold calculated by the probabilistic cumulative distribution function (CDF) value. The experimental results show that our approach improves both the retrieval precision and recall significantly. Ablation studies also illustrate how the probabilistic approach is able to capture the differences between head and tail queries.

## 1 Introduction

Search index is the core technology to enable fast information retrieval based on certain keywords in various modern computational systems, such as web search, e-commerce search, recommendation, and advertising in the past few decades. As a traditional type of search index, inverted index (Dean, 2009), which maps terms to documents in order to retrieve documents by term matching, has been the mainstream type of search index for decades, thanks to its efficiency and straightforward interpretability. Recently, with the advent of the deep learning and pre-training (Schmidhuber, 2015; Kenton & Toutanova, 2019; Liu et al., 2019) era, embedding based retrieval, coupled with approximate nearest neighbor (ANN) search algorithms Johnson et al. (2019), have been established as a promising alternative in search index (Zhang et al., 2020; Huang et al., 2020) and recommendation index (Covington et al., 2016; Zhu et al., 2018; 2019; Gao et al., 2020), in part due to its semantic matching advantage and efficient ANN algorithms.

The origin of using dense vector for retrieval probably can be dated back to Latent Semantic Analysis (LSA) (Deerwester et al., 1990) in the 90s, which is an unsupervised method utilizing singular-value decomposition to get the document representation. Then in the following decades, especially the past decade with the advent of deep learning, the embedding based retrieval model has been increasingly used in information retrieval. Typically, DSSM (Huang et al., 2013), including its variants C-DSSM (Shen et al., 2014), DPSR (Zhang et al., 2020), *etc.* are typical supervised two-tower models, where query and item are embedded into dense vectors in the same semantic space. Then relevant items are retrieved by calculating the cosine similarity between query vectors and item vectors. On top of the above foundational models, recently there are a few more trends going on in the industry: 1) joint training of two-tower model and embedding index (Zhang et al., 2021; Zhan et al., 2021) to avoid the quantization distortion in ANN algorithms. 2) Utilizing large scale of unlabeled data for pre-training (Qiu et al., 2022; Zou et al., 2022) before using user feedback data to train the two-tower model. 3) Apart from embedding index, some methods such as TDM (Zhu et al., 2018), JTM (Zhu et al., 2019) and DR (Gao et al., 2020), utilize a tree-based or other special index structure to introduce a more complex representation model to improve the retrieval precision.

Apart from the architecture innovation, embedding retrieval performs well in various tasks from the view of industrial application. For example, in query parsing tasks, some researchers focus on enhancing query rewriting performance by improving the semantic representation of queries (Li et al., 2022; Qiu et al., 2021), while others utilize retrieval augmentation to facilitate query intent classification (Zhao et al., 2023). In personalized retrieval, user profiles and behaviors are collected to be embed into query representation to facilitate personalized item retrieval (Huang et al., 2020; Li et al., 2021), coupled with technologies like graph learning (Fan et al., 2022; Li et al., 2023), contrastive learning (Dai et al., 2023), *etc.* In multimodal retrieval, researchers mainly focus on solving the semantic gap between the representations of different modals like text and visual resources, facilitating better product search (Zhou et al., 2023) or video search (Wang et al., 2022) in practice.

However, we notice that all the existing works on embedding based retrieval for search and recommendation do not pay enough attention to the two tower model training algorithm, especially the loss function, which in practice causes some tricky problems as follows: 1) the retrieval cutoff is purely a heuristic. Typically, current practical retrieval systems either retrieve a fixed number of items or cut off by a fixed cosine threshold. This heuristic is actually very problematic if we take a closer look. For instance, "iPhone 14" is a very specific query that may only have a handful of relevant products. While "gift" is a very general query that have many relevant products from various categories such as foods, clothes, cosmetics and so on. Thus, one can easily find that retrieving a fixed number of items would not be the optimal solution here, since if the number is small we will miss many relevant products for "gift" and if the number is large we will introduce many irrelevant items for "iPhone 14". On the other hand, it is also sub-optimal to use a fixed cosine threshold, since all the existing model training algorithms have never taken that into account in their design. To the best of our knowledge, there is no satisfactory solution to address this challenge in the industry. 2) Each query is treated equally, no matter they are head, torso or tail queries. However it is well known that queries with different popularity levels have very different performances in a real world search system (Zhang et al., 2020). Thus, it is worth to take the popularity into account when building the model. 3) There is no underlying probabilistic distribution assumption for the relevant items given a query, thus the model falls short of generalization ability. In particular for long-tail queries, it is hard to learn the item distribution due to insufficient training data, leading to suboptimal retrieval performance.

Before we introduce our new approach to address the above shortcomings, let's first review the previous works that inspired our solution: probabilistic models, or more precisely probabilistic graphical models, which have a long history of successes in machine learning (Murphy, 2012; Koller & Friedman, 2009). There are a few models that significantly changed the landscape of machine learning in the past a few decades. Typically, Hidden Markov Model (HMM) (Peña et al., 2001) and the associated Viterbi algorithm (Forney, 1973) for inference and the Baum-Welch algorithm for training (McCallum, 2004), are widely used for modeling sequential data in tasks such as speech recognition, bio-informatics, and natural language processing. Later, Unlike HMM defined in a directional graph, Conditional Random Field (CRF) (Lafferty et al., 2001) defined in undirectional graph gains more popularity in structured prediction tasks such as named entity recognition, information extraction and so on. Later, Latent Dirichlet Allocation (LDA) (Blei et al., 2003), also known as topic modeling which considers each topic as a probability distribution over a fixed vocabulary of words, becomes one of the most successful probabilistic models in machine learning, mostly because of its scalability to larger dataset thanks to the variational inference techniques. While these models are elegant in theory and have proven successful in small data set, they often face challenges when it comes to scalability. In recent years, benefiting by the advancements in computational power, algorithms, and data availability, there has been growing interest in combining probabilistic modeling with neural network. This integration aims to enhance the capability of probabilistic models in various domains, including neural language processing tasks (Tian et al., 2014; Nguyen et al., 2017) and computer vision tasks (Malinin, 2019; Kohl et al., 2018).

Inspired by the previous successes of probabilistic spirits, here we propose a redesign of the embedding based retrieval to address the above challenges by using probabilistic modeling. Our contributions and the overview of the following sections can be summarized as follows.

- First, we introduce the standard two-tower model in Section 2.1, and the widely used frequentist loss functions, point-wise in Section 2.2.1 and pair-wise in Section 2.2.2.

- Then, we propose our two types of loss functions: maximum likelihood estimation based loss in Section 2.3 and contrastive estimation based loss; the latter includes two instances, ExpNCE in Section 2.4.1 and BetaNCE in Section 2.4.2.
- Finally, we conduct comparative experiments to demonstrate the effectiveness of our model in Section 3.2 and ablation study to understand how the model works in Section 3.3.
- To the best of our knowledge, our paper is the first to introduce probabilistic modeling into the embedding based retrieval.

## 2 METHOD

In this section, we first formulate the embedding retrieval problem based on a two-tower architecture and review the existing frequentist approaches. Next, we explore the probabilistic approach which contains the choice of item distribution function and our carefully designed loss function based on the item distribution.

### 2.1 PROBLEM DEFINITION AND NOTATIONS

The embedding retrieval model is typically composed of a query (or a user) tower and an item tower. For a given query $q$ and an item $d$, the scoring output of the model is

$$f(q, d) = s(\mathbf{v}_q, \mathbf{v}_d),$$

where $\mathbf{v}_q \in \mathbb{R}^n$ denotes the $n$-dimensional embedding output of the query tower. Similarly, $\mathbf{v}_d \in \mathbb{R}^n$ denotes the embedding output of item tower, of the same dimension to facilitate efficient retrieval. The scoring function $s(.,.)$ computes the final score between the query and item. Researchers and practitioners often choose $s$ to be the cosine similarity function, equivalently inner product between normalized vectors, *i.e.*, $s(\mathbf{v}_q, \mathbf{v}_d) = \mathbf{v}_q^\top \mathbf{v}_d$, where the superscript $^\top$ denotes matrix transpose. This simple setup has proven to be successful in many applications, *e.g.* (Covington et al., 2016). The objective is to select the top $k$ items $d_1, \ldots, d_k$ from a pool of candidate items for each given query $\mathbf{v}_q$, in terms of $s(\mathbf{v}_q, \mathbf{v}_{d_i})$.

The key design principle for such two-tower architecture is to make the query embedding and the item embeddings independent of each other, thus we can compute them separately after the model is trained. All item embeddings can be computed offline in order to build an item embedding index for fast nearest neighbor search online (Johnson et al., 2019), and the query embedding can be computed online to handle all possible user queries. Even though the embeddings are computed separately, due to the simple dot product interaction between query and item towers, the query and item embeddings are still theoretically in the same geometric space. Then a fixed threshold of item number is used to get the top $k$ relevant items. Thus, typically, the goal is equivalent to minimizing the loss for $k$ query item pairs where the query is given.

### 2.2 EXISTING FREQUENTIST APPROACHES

Most existing methods follow the frequentist approach to the loss function designs for embedding retrieval, which we can categorize into two typical cases: point-wise loss, and pair-wise loss. In the following section, we discuss each of them.

#### 2.2.1 POINT-WISE LOSS

The point-wise based method converts the original retrieval task into a binary classification, where each pair of query and item is computed individually. The goal is to optimize the embedding space where the similarity between the query and its relevant item is maximized, while the similarity between the query and irrelevant item is minimized. It usually adopts the the sigmoid cross entropy loss to train the model. The loss function can be defined as below

$$L_{\text{pointwise}}(\mathcal{D}) = -\sum_i \log \text{sigmoid}(s(\mathbf{v}_{q_i}, \mathbf{v}_{d_i^+})) + \sum_{ij} \log \text{sigmoid}(s(\mathbf{v}_{q_i}, \mathbf{v}_{d_{ij}^-})),$$

where $d_i^+$ denotes the relevant items for query $q_i$ and $d_{ij}^-$ denotes the irrelevant ones, $\text{sigmoid}(x) = 1/(1+\exp(-x))$ is the standard sigmoid function. Though rarely seen in literature, this loss function works well in practice.

### 2.2.2 PAIR-WISE LOSS

This kind of method aims to learn the partial order relationship between positive items and negative items from the perspective of metric learning, closing the distance between the query and positive item and pushing away from the negative. The classical work contains triple loss, margin loss, A-Softmax loss, and several variants (margin angle cosine). Without loss of generality, we formulate the loss as softmax cross-entropy

$$L_{\text{pairwise}}(\mathcal{D}) = -\sum_i \log \text{softmax}(s(\mathbf{v}_{q_i}, \mathbf{v}_{d_i^+}), \{s(\mathbf{v}_{q_i}, \mathbf{v}_{d_{ij}})\}_j)$$

$$= -\sum_i \log \left( \frac{\exp(s(\mathbf{v}_{q_i}, \mathbf{v}_{d_i^+})/\tau)}{\sum_j \exp(s(\mathbf{v}_{q_i}, \mathbf{v}_{d_{ij}})/\tau)} \right),$$

where $\tau$ is the so-called temperature parameter: lower temperature is less forgiving of mis-prediction of positive items by the model. In the same direction, researchers later proposed more advanced loss functions, by introducing max margin in angle space (Liu et al., 2017), in cosine space (Wang et al., 2018) and so on.

### 2.2.3 LIMITATIONS

Both point-wise and pair-wise loss functions have their advantages and limitations. Point-wise loss functions are relatively simpler to optimize, while they may not capture the partial order relationships effectively. In contrast, Pair-wise loss functions explicitly consider the relative ranking between items. As a result, pair-wise loss functions usually achieve better performance in embedding retrieval tasks. While both of them are frequentist approaches, in the sense that no underlying probabilistic distribution are learned and consequently there is no cutoff threshold in principle when retrieving items. Thus, we propose the following probabilistic approach to a more theoretically well founded retrieval.

### 2.3 RETRIEVAL EMBEDDINGS AS A MAXIMUM LIKELIHOOD ESTIMATOR

Given a query $q$, we propose to model the probability of retrieving item $d$, i.e. $p(d|q)$, which is related to the relevance between query $q$ and item $d$, thus

$$p(d|q) \propto p(r_{d,q}|q),$$

where $r_{d,q}$ represents the relevance between the query $q$ and document $d$. For all relevant items $\mathbf{d}^+$, we assume $r_{d^+,q}$, where $d^+ \in \mathbf{d}^+$, follows a distribution whose probability density function is $f_\theta$. For all irrelevant items $\mathbf{d}^-$, we assume $r_{d^-,q}$, where $d^- \in \mathbf{d}^-$, follows a distribution of whose probability density function is $h_\theta$. The likelihood function can be defined as

$$L(\theta) = \prod_q \left( \prod_{\mathbf{d}^+} p(d^+|q) \prod_{\mathbf{d}^-} p(d^-|q) \right)$$

$$\propto \prod_q \left( \prod_{\mathbf{d}^+} p(r_{d^+,q}|q) \prod_{\mathbf{d}^-} p(r_{d^-,q}|q) \right)$$

$$= \prod_q \left( \prod_{\mathbf{d}^+} f_{\theta,q}(r_{d^+,q}) \prod_{\mathbf{d}^-} h_{\theta,q}(r_{d^-,q}) \right).$$

Importantly, the density for relevant ($f_{\theta,q}$) and irrelevant ($h_{\theta,q}$) items are both query dependent. This is a useful generalization from fixed density since different queries have different semantic scopes. Finally the objective function can be defined as the log-likelihood

$$l(\theta) = \log L(\theta)$$

$$= \sum_q \left( \sum_{\mathbf{d}^+} \log f_{\theta,q}(r_{d^+,q}) + \sum_{\mathbf{d}^-} \log h_{\theta,q}(r_{d^-,q}) \right).$$

When we choose different distributions for relevant and irrelevant items, the loss function can resemble a point-wise loss, which may lead to suboptimal performance compared to pair-wise loss functions. To address the limitation, we propose new probabilistic loss functions based on the principles of contrastive loss, which is a well-known pair-wise loss function.

## 2.4 RETRIEVAL EMBEDDINGS AS A NOISE CONTRASTIVE ESTIMATOR

Apart from the above maximum likelihood estimator, the most widely used technique for retrieval model optimization is based on the InfoNCE loss (Oord et al., 2018), which is a form of noise contrastive estimator (an alternative to the maximal likelihood estimator) of the model parameter. In that setting, we need to choose two distributions, the so-called positive sample distribution $p(d^+|q)$, and background (noise) proposal distribution $p(d)$. In theory, the two are related, if we know the joint distribution of $d$ and $q$. But in practice, we can either treat them as separate or simply hypothesize their ratio as the scoring function $r(d, q) := p(d|q)/p(d)$, without knowing them individually. The loss we are minimizing is thus the following negative log-likelihood of correctly identifying the positive item within the noise pool

$$L_r = -\sum_i \log \frac{r(d_i^+, q_i)}{\sum_j r(d_{ij}, q_i)}. \tag{1}$$

By minimizing the loss function, the model aims to maximize $p(d^+|q)$ for the query $q$ and one of its relevant items $d^+$, while pushing away $q$ and its irrelevant items $d_j^+$, thus assign higher similarities to relevant items compared to irrelevant items. Based on the definition, in this section, we propose two types of distributions as the basis of the estimator, truncated exponential distribution in Section 2.4.1 and Beta distribution in Section 2.4.2.

### 2.4.1 PARAMETRIC EXPONENTIAL INFONCE (EXPNCE)

Here we propose to use the following truncated exponential distribution density function as the scoring function $r(d, q) \propto \exp(\cos(\mathbf{v}_d, \mathbf{v}_q)/\tau_q)$ where the function $\cos$ stands for the cosine similarity between the two vectors and the temperature $\tau_q$ is query dependent. This offers an interesting alternative to the standard InfoNCE loss with log bi-linear distribution as

$$L_{\text{ExpNCE}} = \sum_i \log \left( 1 + \frac{\sum_j \exp(\cos(\mathbf{v}_{q_i}, \mathbf{v}_{d_{ij}^-})/\tau_q)}{\exp(\cos(\mathbf{v}_{q_i}, \mathbf{v}_{d_i^+})/\tau_q)} \right).$$

A nice property with the above probabilistic modeling is that we can use a simple cumulative density function (CDF) to decide the cutoff threshold. The CDF for the above truncated exponential distribution can be derived as follows

$$\mathbb{P}_{\text{ExpNCE}}(x < t) = \frac{\int_{-1}^t e^{x/\tau_q} dx}{\int_{-1}^1 e^{x/\tau_q} dx} = \frac{e^{t/\tau_q} - e^{-1/\tau_q}}{e^{1/\tau_q} - e^{-1/\tau_q}},$$

which can be easily computed.

### 2.4.2 PARAMETRIC BETA INFONCE (BETANCE)

We call a probability distribution compactly supported if its cumulative distribution function $F$ satisfies $F((-\infty, -x]) = 1 - F([x, \infty)) = 0$ for some $x > 0$. In other words, all the mass is contained in the finite interval $[-x, x]$. The best-known family of compact distributions in statistics is probably the Beta distributions, whose pdf are defined on $[0, 1]$. Since the cosine similarity used in two-tower model has value range of $[-1, 1]$, we expand the definition range of Beta distributions to $[-1, 1]$ and define its density as

$$f_{\alpha,\beta}(x) = \frac{\Gamma(\alpha + \beta)}{2\Gamma(\alpha)\Gamma(\beta)} \left( \frac{1+x}{2} \right)^{\alpha-1} \left( \frac{1-x}{2} \right)^{\beta-1}.$$

An interesting special case is when $\alpha = \beta = 1$, which gives the following uniform CDF on $[-1, 1]$

$$F_{1,1}(x) = \frac{x+1}{2}.$$

One difficulty working with Beta distributions is that its CDF has no closed form, but rather is given by the incomplete Beta function [1]: $B(t; \alpha, \beta) = \int_0^t x^{\alpha-1}(1-x)^{\beta-1}dx$ is the incomplete Beta integral. The CDF for the above Beta distribution can be derived as follows.

$$\mathbb{P}_{\text{BetaNCE}}(x < t) = \frac{\int_{-1}^{t}((1+x)/2)^{\alpha-1}((1-x)/2)^{\beta-1}dx}{\int_{-1}^{1}((1+x)/2)^{\alpha-1}((1-x)/2)^{\beta-1}dx}$$

$$= \frac{\int_0^{\frac{t+1}{2}} x^{\alpha-1}(1-x)^{\beta-1}dx}{\int_0^1 x^{\alpha-1}(1-x)^{\beta-1}dx}$$

$$= \frac{\Gamma(\alpha+\beta)}{\Gamma(\alpha)\Gamma(\beta)}B(\frac{t+1}{2}; \alpha, \beta).$$

Fortunately we only need the PDF during training; the CDF is needed only once, to determine the retrieval threshold after training (see Appendix A). Then we assume $p(d|q)$ and $p(d)$ follow beta distributions like

$$p(d|q) = g_q(s) \propto z^{\alpha_g(q)}(1-z))^{\beta_g(q)}$$

and

$$p(d) = k_q(s) \propto z^{\alpha_k(q)}(1-z))^{\beta_k(q)},$$

where $z = \frac{1+s}{2}$, s is the cosine similarity between the query and item, $\alpha_g(q)$ and $\beta_g(q)$ are encoders based on the query representation to distinguish the learning of item distribution, and so do $\alpha_k(q)$ and $\beta_k(q)$. According to the definition of $r(d, q)$ in Section 2.4, we can get

$$r(d, q) = \frac{g_q(s)}{k_q(s)} \propto z^{\alpha_g(q)-\alpha_k(q)}(1-z)^{\beta_g(q)-\beta_k(q)},$$

which is again a Beta density. Note that $\alpha, \beta$ correspond to the number of successes and failures that lead to the Beta distribution. We can view both $p(d|q)$ and $p(d)$ as having the same number of failures (being a negative document for $q$), but $p(d|q)$ has some additional successes. Thus we can hypothesize that $\beta_g = \beta_k$, and therefore get the final BetaNCE loss of log-likelihood in Equation (1) as

$$L_{\text{BetaNCE}} = -\sum_i \log\left(\frac{z(\mathbf{v}_{q_i}, \mathbf{v}_{d_i^+})^{\alpha_g(q)-\alpha_k(q)}}{\sum_j z(\mathbf{v}_{q_i}, \mathbf{v}_{d_{ij}})^{\alpha_g(q)-\alpha_k(q)}}\right)$$

$$= -\sum_i \log\left(\frac{e^{\log z(\mathbf{v}_{q_i}, \mathbf{v}_{d_i^+})/\tau_q}}{\sum_j e^{\log z(\mathbf{v}_{q_i}, \mathbf{v}_{d_{ij}})/\tau_q}}\right),$$

where $\tau_q = (\alpha_g(q) - \alpha_k(q))^{-1}$. Thus compared with InfoNCE, the main difference is the logarithmic transformation applied to the cosine similarity. Since the cosine similarity $s$ is empirically always bounded away from $-1$, the logarithm presents no numerical difficulty.

After the training, we can get back the Beta distribution parameters from the learned $\tau_q$ by fixing the background distribution parameters $\alpha_k(q) \equiv 1$, $\beta_k(q) \equiv 1$, that is, the uniform distribution on the unit interval:

$$\alpha_g(q) = \tau_q^{-1} \text{ and } \beta_g(q) = 1. \tag{2}$$

## 3 EXPERIMENTS

In this section, we first explain the details of the dataset, experimental setup, baseline methods and evaluation metrics. Then we show the comparison of experimental results between the baseline methods and pEBR. Finally we show the ablation study to intuitively illustrate how pEBR could perform better.

---

[1] https://www.tensorflow.org/api_docs/python/tf/math/betainc

## 3.1 Experimental Setup

### 3.1.1 Dataset and setup

Our model was trained on a randomly sampled dataset consisting of 87 million user click logs collected over a period of 60 days. We trained the model on an Nvidia GPU A100 and employed the AdaGrad optimizer with a learning rate of 0.05, batch size of 1024, and an embedding dimension of 128. The training process converged after approximately 200,000 steps, which took around 12 hours.

### 3.1.2 Methods in Comparative Experiments

Our method is compatible with most two-tower retrieval models since it primarily modifies the loss function rather than the model architecture. Thus, we choose a classic two-tower retrieval model DSSM (Huang et al., 2013), which has been proven successful in various retrieval scenarios, as the backbone model. Then we focus on cutting off items retrieved by DSSM model with three methods:

- DSSM-topk refers to the method which cuts off retrieved items using a fixed threshold of numbers. In other words, we take the top $k$ items with the highest relevance scores. Note that in this case, all queries have the same number $k$ of retrieved items.
- DSSM-score refers to the method that cuts off retrieved items with a fixed threshold of relevance score. Items with relevance scores below the threshold are discarded. Thus, the queries have varying number of retrieved items. In practice, we also have a maximum number to prevent some head queries to retrieve too many items that the downstream ranking system can not handle.
- pEBR refers to our proposed probabilistic method which cuts off retrieved items with a threshold derived by probabilistic model, specifically the CDF value of the learned item distribution. We are using a default CDF value $0.5$ if not further specified. In this experiment, we focus on using BetaNCE with $\beta = 1$ as shown in Equation (2) to demonstrate the effectiveness of the methods. But our method is general enough to accommodate other distributions, such as the proposed ExpNCE.

Both DSSM-topk and DSSM-score serve as baseline methods in our work, against which we compare the performance of our proposed pEBR method.

### 3.1.3 Experimental metrics

We use two widely reported metrics, Recall@$k$ and Precision@$k$ to evaluate the quality and effectiveness of retrieval models (Zhang et al., 2021; Zhu et al., 2018). Recall@$k$ (R@$k$) is a standard retrieval quality metric that measures the ratio of retrieved relevant items to the total number of relevant items. Precision@$k$ (P@$k$) is another commonly used retrieval metric that calculates the ratio of true relevant items to the total number of retrieved items within the top $k$ results.

There are some nuances that need a little more explanation. In DSSM-topk, a fixed number $k$ of items is retrieved for each query to calculate R@$k$ and P@$k$. While in DSSM-score and pEBR, a threshold of relevance score or a CDF value are used to cut off retrieved items, which results in a different number of items for each query. Thus for fair comparison, we tune the threshold slightly to make a average number $k$ of items to be retrieved for all queries to calculate the metrics.

Moreover, Note that there is a tradeoff between precision and recalls when varying the $k$ value. In the experiments, we choose $k = 1500$ to optimize for recall, since it is the main goal of a retrieval system. Thus, the precision value is relatively very low.

## 3.2 Experimental Results

In this section, we compare the performance of our proposed method with the baseline methods. Since the retrieval performance differs significantly from head, torso to tail queries, we separate our evaluation dataset into three according parts to assess how the proposed method performs.

As shown in Table 1 we can draw the following conclusions: 1) DSSM-score performs better than DSSM-topk on both R@1500 and P@1500, which is as expected since the retrieved items of DSSM-

Table 1: Comparison of retrieval quality between our proposed method and baseline methods.

| | All Queries | | Head Queries | | Torso Queries | | Tail Queries | |
|---|---|---|---|---|---|---|---|---|
| | P@1500 | R@1500 | P@1500 | R@1500 | P@1500 | R@1500 | P@1500 | R@1500 |
| DSSM-topk | 0.327% | 93.29% | 0.782% | 90.38% | 0.211% | 94.36% | 0.126% | 94.36% |
| DSSM-score | 0.435% | 93.64% | 0.841% | 90.77% | 0.339% | 94.64% | 0.275% | 94.57% |
| pEBR | **0.583**% | **94.08**% | **0.945**% | **91.42**% | **0.513**% | **95.00**% | **0.450**% | **94.73**% |

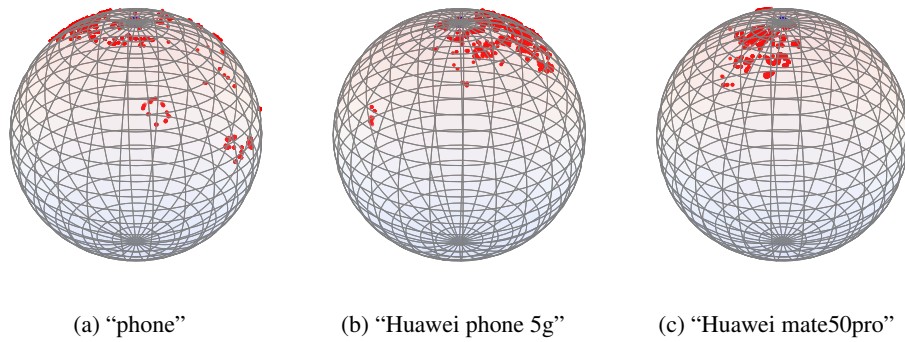

(a) "phone"      (b) "Huawei phone 5g"      (c) "Huawei mate50pro"

Figure 1: Relevant item distributions projected to a sphere for three queries in head, torso, and tail categories, respectively.

score have better relevance scores than DSSM-topk, because items with lower relevance scores are cut off. 2) Our proposed model pEBR outperforms both DSSM-topk and DSSM-score with improvements of 0.79% and 0.44% on R@1500 respectively, which proves that pEBR captures the difference between queries successfully. pEBR learns varying item distributions for different queries, allowing for the determination of dynamic and optimal relevance score thresholds, thus leading to enhanced overall retrieval performance. 3) pEBR achieves better performance on separated evaluation sets, *i.e.* head queries, torso queries and tail queries. Moreover, we observe that the amount of improvement achieved by pEBR varies systematically with respect to query popularities. Specifically, compared to DSSM-topk, pEBR achieves 1.04%, 0.64% and 0.37% recall improvements on head queries, torso queries and tail queries respectively. This is because that head queries normally have much more relevant items than tail queries, thus a dynamic cutoff threshold could benefit head queries to retrieve more items significantly. 4) Similarly, pEBR achieves 0.163%, 0.302% and 0.324% precision improvements on head, torso and tail queries on top of DSSM-topk. This is because that tail queries normally do not have many relevant items, typically just tens of items. Thus retrieving a fixed number $k$ of items hurts the precision significantly. pEBR appears to solve this problem nicely.

## 3.3 ABLATION STUDY

### 3.3.1 VISUAL ILLUSTRATION OF DISTRIBUTIONS

As shown in Figure 1, we visualize the relevant item distribution for three queries with different frequencies. Since it is difficult to visualize 128 embedding dimension, we first need to apply dimension reduction technique, specifically t-SNE (Van der Maaten & Hinton, 2008) here to reduce the dimension to 3-D. Then we normalize the vectors as unit vector and plot them on a sphere. Note that we are using cosine similarity thus the norms of the vectors do not matter. The head query, "phone", is a quite general one that can retrieve phones of various brands and models. As a result, the retrieved items are widely distributed and dispersed across the surface of the sphere. The torso query "Huawei phone 5g" is a more specific one as it focuses on phones from the brand Huawei and with 5G capability. Consequently, the item distribution is more concentrated and has a narrower scope compared to the query "phone". The tail query, "Huawei mate50pro", is highly specific query as it specifies the brand (Huawei) and model (mate50pro), thus the number of relevant items is very small and the distribution becomes even more concentrated. These differences in item distributions

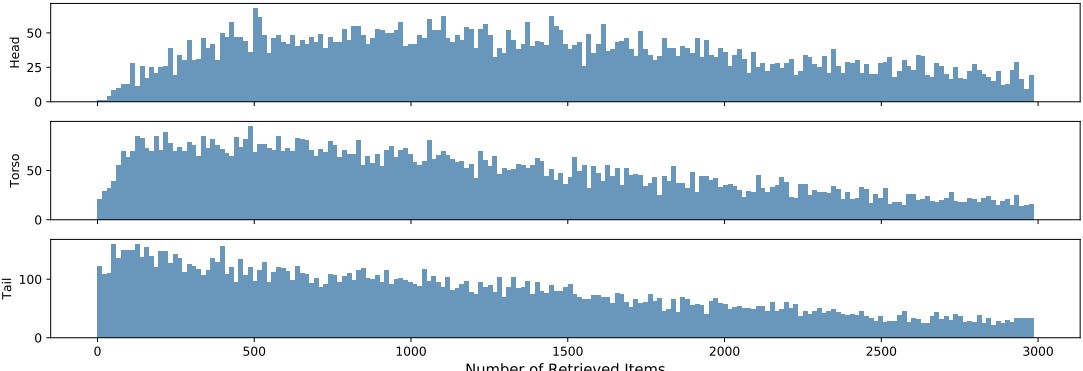

Figure 2: Histogram of the number of retrieved items for head, torso, and tail queries, where we fix the CDF value to be 0.985.

Table 2: Average numbers of retrieved items for different cutoff CDF values.

| Cutoff CDF Value | 0.99 | 0.95 | 0.9 | 0.8 | 0.7 | 0.6 | 0.5 | 0.4 |
|---|---|---|---|---|---|---|---|---|
| Head Queries | 2253.12 | 1443.51 | 1073.79 | 761.06 | 600.31 | 441.51 | 227.42 | 66.36 |
| Torso Queries | 1897.53 | 961.90 | 617.27 | 334.10 | 200.17 | 111.28 | 48.37 | 12.89 |
| Tail Queries | 1804.80 | 813.24 | 473.40 | 215.50 | 113.99 | 58.86 | 21.46 | 4.60 |

reaffirm the conclusion that cutting off retrieved items by a fixed threshold of item numbers or a fixed relevance score is suboptimal for embedding retrieval.

### 3.3.2 Dynamic Retrieval Effect

In Figure 2, we show the histogram of retrieved item numbers cut off by CDF value 0.985 for both head, torso and tail queries. In detail, we filter out the items with the cosine similarity $x$ bellowing a threshold $t$ to ensure the equation $\mathbb{P}(x >= t) = 0.985$. In general, head queries have a more uniform distribution and the numbers lie mostly in the range of [500, 1500], while torso and tail queries share similar skewed distribution and the numbers lie in the range of [0, 1000] and [0, 500], respectively. This indicates that head queries retrieve more items than torso and tail queries, which confirms the assumption that queries with higher popularity need more candidates and queries with lower popularity need fewer.

In Table 2, we present the average number of retrieved items under different CDF values with the same cutting off strategy in Figure 2. As the probability decreases, queries claim a higher relevance level for items, thus the average numbers of retrieved items decreases accordingly. Comparing different queries, head queries retrieve more items than torso and tail queries under all CDF thresholds, which again confirms the conclusion above.

### 4 Conclusion

In this paper, we have proposed a novel probabilistic embedding based retrieval model, namely pEBR, to address the challenges of insufficient retrieval for head queries and irrelevant retrieval for tail queries. Based on the noise constrastive estimator, we have proposed two instance models: ExpNCE and BetaNCE, with the assumption that relative items follow truncated exponential distribution or beta distribution, which allow us to easily compute a probabilistic CDF threshold instead of relying on fixed thresholds, such as a fixed number of items or a fixed relevance score. Comprehensive experiments and ablation studies show that the proposed method not only achieves better performance in terms of recall and precision metrics, but also present desirable item distribution and number of retrieved items for head and tail queries.

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

# A   INFERENCE TRUNCATION

The major goal of using query dependent item distribution in neural retrieval is to give statistical meaning to the retrieved candidate set. Previously a somewhat arbitrary combination of cosine similarity threshold and top-K threshold are used.

$$S_{t,K}(q) = \text{Top}_K^{\cos(q,\cdot)}\left(\{d_i : \cos(q, d_i) \geq t\}\right)$$

The cosine threshold does not account for the variability of item distributions across different queries, while the top-K threshold is mainly to save inference cost.

Now for a given $q$-isotropic spherical distribution $H$, whose marginal density with respect to $\cos(q, \cdot)$ is given by $h : [-1, 1] \to \mathbb{R}_+$, we can compute its CDF as follows

$$\mathbb{P}(H < t) = \frac{\int_{-1}^{t} h(x)(1 - x^2)^{(n-3)/2} dx}{\int_{-1}^{1} h(x)(1 - x^2)^{(n-3)/2} dx}. \tag{3}$$

For a given $h$ and $t$, (3) can be computed numerically. For the two special $h$ we are concerned here, we can give semi-closed forms:

- For the Beta density (BetaNCE) $h(x) \propto (1+x)^{\alpha-1}(1-x)^{\beta-1}$, the normalization constant, after the rescaling $[-1, 1] \to [0, 1]$, is a complete Beta integral

$$\int_0^1 x^{\alpha + \frac{n-5}{2}}(1 - x)^{\beta + \frac{n-5}{2}} dx = B(\alpha + \frac{n-3}{2}, \beta + \frac{n-3}{2})$$
$$= \frac{\Gamma(\alpha + \frac{n-5}{2})\Gamma(\beta + \frac{n-5}{2})}{\Gamma(\alpha + \beta + n - 4)},$$

  while the numerator is proportional to an incomplete Beta integral

$$\int_0^{\frac{1+t}{2}} x^{\alpha + \frac{n-5}{2}}(1 - x)^{\beta + \frac{n-5}{2}} dx =: B_{\frac{1+t}{2}}(\alpha + \frac{n-3}{2}, \beta + \frac{n-3}{2}).$$

- For the truncated exponential density (corresponding to InfoNCE), $h(x) \propto e^{x/\tau}$, the integral we need to compute is the following modified Bessel integral

$$\int_{-1}^{t} e^{x/\tau}(1 - x^2)^{\frac{n-3}{2}} dx = \int_{-1}^{t} e^{x/\tau}(1 + x)^{\frac{n-1}{2} - 1}(1 - x)^{\frac{n-1}{2} - 1} dx.$$

  This admits a closed form solution for any $t$ provided $n > 2$ is odd, however the solution has alternating signs, which is numerically unstable especially for large $n$.

Due to the difficulty of exact solutions, we resort to numeric quadratures:

```
import scipy.integrate as integrate
import scipy.special as special
import math

def BetaInt(alpha, beta, t):
    return integrate.quad(lambda x: (1 + x) ** (alpha - 1) * \
    (1 - x) ** (beta - 1), -1, t)

def BetaExpInt(alpha, beta, tau, t):
    return integrate.quad(lambda x: (1 + x) ** (alpha - 1) * \
    (1 - x) ** (beta - 1) * math.exp(x / tau), -1, t)

cache = {}

def CosineInvCDF(p, quad_fn, *args):
```

```
    values = cache.setdefault(quad_fn, {}).get(tuple(args))
    if not values:
        denom = quad_fn(*args, 1)[0]
        values = cache[quad_fn][tuple(args)] = \
        [quad_fn(*args, i / 500 - 1)[0] / denom for i in range(1001)]
    if p >= values[-1]:
        return 1
    if p <= values[0]:
        return -1
    right = min(i for i, v in enumerate(values) if v >= p)
    left = right - 1
    return (right * (p - values[left]) + left * (values[right] - p)) / \
    (values[right] - values[left]) / 500 - 1

def BetaInvCDF(p, alpha, beta):
    return CosineInvCDF(p, BetaInt, alpha, beta)

def BetaExpInvCDF(p, alpha, beta, tau):
    return CosineInvCDF(p, BetaExpInt, alpha, beta, tau)

def InfoNCEInvCDF(p, n, tau):
    return CosineInvCDF(p, BetaExpInt, (n-1)/2, (n-1)/2, tau)
```

