# OpenReview forum: "pEBR: A Probabilistic Approach to Embedding Based Retrieval"
_ICLR.cc/2024/Conference — Submitted to ICLR 2024_

### Official Review · Reviewer_xamY · 2023-11-01

**Soundness:** 2 fair
**Presentation:** 3 good
**Contribution:** 2 fair
**Rating:** 5
**Confidence:** 3

**Summary:**

This paper considers a MLE rather than frequentist approach to training retrieval embeddings. The model posits a PDF for the relevance of a document to a query based on the inner product of the representation of the query and the document (Beta distribution). Subsequent optimization for the model parameters yields the vector representation. The dataset used contains 87M clicks and the result improves over the DSSM model on both precision (by small largin) and recall (by a bigger margin, especially on tail data).

**Strengths:**

1. Improved precision compared to DSSM, especially on tail queries. Recall is better than DSSM but by a small margin.
2. The model produces a variable number of results based on relevance cutoff. So where there are more results relevant to a query, the model can retrieve more of them compared to DSSM.

**Weaknesses:**

1. Assessment is sparse. The baseline chosen, DSSM, is rather old (from 2013).
2. Comparison on one dataset and its unclear if it is public.

**Questions:**

1. Can you comment on the dependence on the amount of training data? Would the precision and recall be higher with much smaller click data than 87M clicks?
2. Why not compare to SoTa model for retrieval?
3. Have you tried evaluated it on other public datsets?

---

> ### Author Response · Authors · 2023-11-19
> **Point to Point Responses**
>
> Thanks very much for the valuable comments. Below are our point to point responses:
>
> Responses to the weaknesses:
> 1. Since our proposed method is applicable to various types of networks, thus we choose the classical DSSM model as a representative to present the model performance.
> 2. It is the real data in our online e-commerce system, we consider to publish it in the future.
>
> Responses to the questions:
> 1. In general, larger dataset will result in better performance. However, our method has advantages with smaller datasets since we introduce probabilistic distribution assumption to improve the generalization ability.
> 2. Since our proposed method is applicable to various types of networks, thus we choose the classical DSSM model as a representative to present the model performance.
> 3. Not yet, but we consider to publish our dataset in the future.

---

### Official Review · Reviewer_vdaa · 2023-11-06

**Soundness:** 2 fair
**Presentation:** 3 good
**Contribution:** 2 fair
**Rating:** 3
**Confidence:** 2

**Summary:**

This work presents a probabilistic framework for performing retrieval with embedding models. The rationale is that standard techniques that usually employ fixed number of items to retrieve or with a tuned score will impact precision and recall metrics for different queries. The authors propose a probabilistic approach in the setting of two-tower approaches, by extending the InfoNCE loss. The approach is evaluated in a dataset of user click logs and compared with two baselines.

**Strengths:**

The work is around dense embedding retrieval which is an important topic specially in industrial applications with large catalogs of items.
It is interesting to see this probabilistic approach for retrieving items as it avoids using standard approaches which may bring inefficiencies.

**Weaknesses:**

It would be great if the authors could enhance the related work with probabilistic embedding approaches especially few from the domain of images and metric learning and also draw some parallels.For example, Probabilistic Embeddings for Cross-Modal Retrieval, CVPR 2021.
One could use such an approach for performing retrieval tasks as it models uncertainty.

Is very difficult to assess the result. Could you please give more details for the dataset? What type of user log are these? How the model that you train looks like? What features do you have? How large is the set of unique items? The dataset as well as the architecture used is described very briefly.

Why you select such a large number for k? Does this artificially inflate the metrics you measure? Usually we would try to retrieve a small set of elements to feed to a ranker.

How significant is the result that you achieve? How this affects the ranking stage?

The experimentation part is very weak and is hard to assess the effectiveness of the approach.

**Questions:**

Please previous comments.

---

> ### Author Response · Authors · 2023-11-19
> **Point to Point Responses**
>
> Thanks very much for the valuable comments. Below are our point to point responses:
>
> 1. We think our work quite different from the probabilistic embedding work presented at CVPR 2021. As the probabilistic embedding is mainly focus on the learning of polysemy, our work aims to learn popularity. Therefore, we didn't delve into a detailed discussion of probabilistic embedding. But it is also a kind of probabilistic method, we will consider to discuss more about these methods in the future.
> 2. The datasets are user click logs collected from a large-scale e-commerce system, which have tens of millions of unique queries and items. The model we employed is based on the DSSM architecture which has a query tower and an item tower. The query tower takes the input of user input query, and the item tower takes the attributes of item like title, brand, model, color, size and so on.
> 3. The choice of $k$ depends on the efficiency requirements of online system. We choose a large $k$ for two reasons: (1) our e-commerce system is large-scale, which have billions of candidate items and we need to retrieval enough items for subsequent ranking stage. (2) We have cascade ranking modules which serves like a funnel, progressively reducing the computation at each stage.
> 4. We have achieve significant improvement in online A/B tests. It retrieves more items for popular queries, while less items for specific queries, thus the total number for ranking stage does not change much.
> 5. About the baseline method, we didn't find any previous work similar to ours, and our proposed method is applicable to various types of networks, thus we choose the classical DSSM model as a representative to present the model performance.

---

> > ### Comment · Reviewer_vdaa · 2023-11-22
> >
> > Thanks for the replies.
> >
> > Still, the details of the system and datasets are unknown to me. Even in your reply everything is very vague. No reproducibility for this paper is possible.
> >
> > No evaluation on public datasets.

---

### Official Review · Reviewer_2BmG · 2023-11-08

**Soundness:** 3 good
**Presentation:** 4 excellent
**Contribution:** 2 fair
**Rating:** 5
**Confidence:** 3

**Summary:**

The paper proposes a probabilistic approach to embedding-based retrieval, which allows for the design of a dynamic retrieval cutoff strategy tailored to different types of queries.

**Strengths:**

The paper is generally well-written with a clear motivation from the weaknesses of existing frameworks. The authors present empirical evidence to the central research problem.

**Weaknesses:**

1. The authors claim that the paper is the first to introduce probabilistic modelling into embedding based retrieval, which remains doubtful to me. Probabilistic embedding has a long history in machine learning, as well as probabilistic information retrieval, at least dating back to probabilistic ranking principle (Robertson, 1977), which essentially seeks to model the relevance of items to a query. Such literature was not reviewed in the paper. Furthermore, this formulation for modelling the retrieval probability and learning embeddings based on contrastive losses is not new to the community (see [1] and the references therein).

2. Given the development of embedding retrieval models, the chosen baseline DSSM is quite old (2013). which cannot substantiate the usefulness of the proposed method over existing approaches.

[1] Chun et al. (2021). Probabilistic Embeddings for Cross-Modal Retrieval.

**Questions:**

1. In Section 3.3.2, the current model pEBR also relies on the chosen threshold $t$. Could the authors explain how sensitive it is to the model performance and at what value one should set it in practice?

2. The choice of fixed threshold for $K$ items is mainly for saving inference cost. As detailed in Appendix A, it seems that computing the probabilistic CDF threshold can be slow, thus practically undesirable given the time constraints. If this system is in place, could the authors verify this has little impact on the product experience?

3. The authors claim that without the probabilistic assumption, the model can fall short of generalisation ability. Any empirical evidence to support this, in comparison with the frequentist approaches?

---

> ### Author Response · Authors · 2023-11-19
> **Point to Point Responses**
>
> Thanks very much for the valuable comments. Below are our responses:
>
> Responses to weakness:
> 1. We know that there have been several literatures about probabilistic embeddings. However, we mainly focus on learning the probabilistic distribution of the relevance score of query and item, which is distinct from the learning of probabilistic embeddings. Furthermore, most probabilistic embeddings aim to capture the polysemy of query or item, our work addresses the learning of popularity.
> 2. About the baseline method, we didn't find any previous work similar to ours, and our proposed method is applicable to various types of networks, thus we choose the classical DSSM model as a representative to present the model performance.
>
> Responses to questions:
> 1. We need to consider both the effectiveness and efficiency of an online system. Larger $t$ brings better retrieval performance but require more online computations. In practice, we conduct several A/B tests to choose different $t$ that can balance the effectiveness and efficiency.
> 2. Yes, the probabilistic CDF threshold requires more computing, but it has not yet being complex enough to impact system efficiency.
> 3. Retrieval system usually suffers from the long-tail problem. The frequentist approaches are unable to effectively learn the probabilistic distribution of long-tail queries, as these queries are always short of training data.

---

> > ### Comment · Reviewer_2BmG · 2023-11-19
> > **Thank you!**
> >
> > Thanks for the responses. Given that the emphasis is on practicality of the system, it seems a common concern among the reviewers that the proposed method needs further comparison against popular methods and even on more public datasets. If the proposed method were superior or at least competitive,  we could truly know how useful it is and I would be willing to increase the score. I highly suggest the authors consider additional experimentation. Otherwise, the proposal is not convincing to us.

---

### Meta-Review · Area_Chair_Hmky · 2023-12-05

**Metareview:**

(a)  The paper proposes a probabilistic approach to embedding-based retrieval, which allows for the design of a dynamic retrieval cutoff strategy tailored to different types of queries.  The paper uses an MLE approach and some theory for this.
(b)  Well written.
(c)    Needs more comparison methods and more public datasets.

**Justification For Why Not Higher Score:**

Consensus on Reject.

**Justification For Why Not Lower Score:**

N/A

---

### Decision · Program_Chairs · 2024-01-16

Reject